# Improving the phenotype predictions of a yeast genome-scale metabolic model by incorporating enzymatic constraints

Benjamín J Sánchez[1,2,†] , Cheng Zhang[3,4,†] , Avlant Nilsson[1] , Petri-Jaan Lahtvee[1,2] , Eduard J Kerkhoven[1,2] & Jens Nielsen[1,2,5,*]

## Abstract

Genome-scale metabolic models (GEMs) are widely used to calculate metabolic phenotypes. They rely on defining a set of constraints, the most common of which is that the production of metabolites and/or growth are limited by the carbon source uptake rate. However, enzyme abundances and kinetics, which act as limitations on metabolic fluxes, are not taken into account. Here, we present GECKO, a method that enhances a GEM to account for enzymes as part of reactions, thereby ensuring that each metabolic flux does not exceed its maximum capacity, equal to the product of the enzyme's abundance and turnover number. We applied GECKO to a *Saccharomyces cerevisiae* GEM and demonstrated that the new model could correctly describe phenotypes that the previous model could not, particularly under high enzymatic pressure conditions, such as yeast growing on different carbon sources in excess, coping with stress, or overexpressing a specific pathway. GECKO also allows to directly integrate quantitative proteomics data; by doing so, we significantly reduced flux variability of the model, in over 60% of metabolic reactions. Additionally, the model gives insight into the distribution of enzyme usage between and within metabolic pathways. The developed method and model are expected to increase the use of model-based design in metabolic engineering.

**Keywords** enzyme kinetics; flux balance analysis; molecular crowding; proteomics; *Saccharomyces cerevisiae*
**Subject Categories** Genome-Scale & Integrative Biology; Metabolism; Methods & Resources
**Mol Syst Biol. (2017) 13: 935**

## Introduction

Metabolism is at the core of cellular function; the development of reliable quantitative models of metabolism is thus a main objective of systems biology. For the past 20 years, a recurrent modeling approach for reaching this objective has been constraint-based modeling (Lewis *et al*, 2012; Palsson, 2015), which enables calculation of metabolic fluxes from reactions' stoichiometry and intracellular metabolites' mass balances. Genome-scale models (GEMs), which are genome-wide constraint-based models, have been used extensively for metabolic engineering applications such as yield or knockout predictions (Kerkhoven *et al*, 2014; O'Brien *et al*, 2015). However, when considering the production of a metabolite of interest, these models typically make the assumption that the uptake rate of the carbon source (e.g., glucose) limits production. This may be an oversimplification, as metabolic fluxes are limited by their corresponding enzyme levels. However, this cannot be directly tested in traditional GEMs because they do not allow for connecting enzyme concentrations to metabolic fluxes. Therefore, there is interest in developing novel modeling concepts that will enable the incorporation of enzyme levels in GEMs, particularly as quantitative proteomics data, from which enzyme levels can be inferred, become more available. Proteomics data have so far mostly been indirectly combined with GEMs by correlating protein levels to the corresponding fluxes (Sánchez & Nielsen, 2015).

Different approaches have been developed to account for enzymatic limitations in metabolic models. One approach, flux balance analysis with molecular crowding (FBAwMC) (Beg *et al*, 2007), relies on imposing a global capacity constraint on the total cellular volume occupied by all metabolic enzymes. The approach has also been adapted to constrain the total mass of the enzymes (Shlomi *et al*, 2011). Using FBAwMC together with a GEM of *Escherichia coli*, it was shown that acetate production at a high specific growth rate is due to the low catalytic efficiency of oxidative phosphorylation (Beg *et al*, 2007; Vazquez *et al*, 2008). Similar results were

---

1 Department of Biology and Biological Engineering, Chalmers University of Technology, Gothenburg, Sweden
2 Novo Nordisk Foundation Center for Biosustainability, Chalmers University of Technology, Gothenburg, Sweden
3 Science for Life Laboratory, KTH – Royal Institute of Technology, Stockholm, Sweden
4 State Key Laboratory of Bioreactor Engineering, East China University of Science and Technology, Shanghai, China
5 Novo Nordisk Foundation Center for Biosustainability, Technical University of Denmark, Hørsholm, Denmark
*Corresponding author. Tel: +46 31 772 3804; E-mail: nielsenj@chalmers.se
†These authors contributed equally to this work as first authors

observed using variations of the approach in lactate-producing cancerous human cells (the Warburg effect) (Shlomi *et al*, 2011; Vazquez & Oltvai, 2011) and in ethanol-producing *Saccharomyces cerevisiae* cells (the Crabtree effect) (Van Hoek & Merks, 2012; Nilsson & Nielsen, 2016). Other variations of the approach have been developed to consider enzymes as a separate entity (Adadi *et al*, 2012) and to account for additional protein sectors (Mori *et al*, 2016). Nonetheless, these approaches were developed to study the global adaptation of the proteome to physiochemical constraints and are hence not designed for the integration of proteomic data.

An alternative framework to account for enzyme limitation is a genome-scale model of both metabolism and gene expression (ME model) (O'Brien & Palsson, 2015), which includes metabolic reactions and all processes required for the synthesis of functional proteins starting from the transcription rates of genes. This approach has been used to confirm a limitation in enzyme capacity at a high specific growth rate in *E. coli* (O'Brien *et al*, 2013). ME models have only been developed for *Thermotoga maritima* (Lerman *et al*, 2012) and *E. coli* (Orth *et al*, 2010) because they require detailed knowledge of all the steps of protein synthesis (protein maturation, protein folding, metal binding, etc.), which are not readily available for all organisms, and in particular for eukaryal cells. Although there have been recent efforts in modeling the protein secretion process in those organisms (Feizi *et al*, 2013; Liu *et al*, 2014b), details of the protein synthesis requirements are needed, especially in terms of localization and compartmentalization.

Alternative approaches such as resource balance analysis (Goelzer *et al*, 2015), self-replicating models (Molenaar *et al*, 2009; Berkhout *et al*, 2013), and whole-cell models (Karr *et al*, 2012) have also been developed to account for protein limitations. The former estimates apparent catalytic rates from experimental data and uses the estimations as hard constraints (equalities) to predict protein distribution; therefore, it requires multiple experimental datasets. The latter two approaches either are mostly qualitative in nature or require an excessive number of parameters that are not currently available. Considering all aforementioned modeling approaches, a quantitative predictive genome-scale method is needed that can impose soft constraints (inequalities) on each enzyme level for integration of proteomics data.

Here, we present a comprehensive modeling approach for using enzyme kinetics and abundances to constrain a GEM to biologically feasible fluxes. In our methodology, each metabolic reaction includes an extra entity that represents enzyme usage. The entity is limited by the protein abundance, which can be provided as input to the model. Thus, we can conveniently simulate metabolism with constraints based on protein abundance measurements, correctly representing capacity constraints on fluxes. The method enhances a *G*EM with *E*nzymatic *C*onstraints using *K*inetic and *O*mics data and is referred to as GECKO. We applied GECKO to a GEM of *S. cerevisiae* and show how different biological phenomena can be explained with the approach. In particular, through simulation, we show that enzyme limitation governs different cellular behaviors, such as gene knockout phenotype, growth on different carbon sources, and yields of secreted metabolites. These results reinforce the idea that there is a simple principle for protein allocation in microorganisms (Basan *et al*, 2015; Hui *et al*, 2015; Nilsson & Nielsen, 2016).

# Results

## GECKO: accounting for enzyme constraints in a genome-scale model

Any reaction flux (or metabolic rate) has a basic constraint: The flux cannot exceed the reaction's maximum rate ($v_{max}$), which is equal to the intracellular concentration of the corresponding enzyme multiplied by the enzyme's turnover number ($k_{cat}$ value). However, intricate relationships between enzymes and reactions are quite frequent and complicate the aforementioned constraint (Adadi *et al*, 2012). Examples of this include isozymes, i.e., different enzymes that catalyze the same reaction; promiscuous enzymes, which can catalyze different reactions; complexes, in which several subunits together catalyze one reaction; and reversible reactions, where an enzyme catalyzes both directions of the same reaction.

We developed GECKO to limit metabolic fluxes in any GEM with enzymatic data in a simple manner, so we can reduce the variability of constraint-based modeling results and improve predictions. The approach extends genome-scale modeling by representing enzymes as entities with limited capacities in the corresponding reactions in the model (Fig 1A). In traditional genome-scale modeling, a stoichiometric matrix representing the whole metabolism is defined in which columns indicate each reaction's stoichiometry, and rows indicate the mass balance for each metabolite. In GECKO, we expanded the approach by adding new rows to this matrix that represent the enzymes and new columns that represent each enzyme's usage (Fig 1B; Gu *et al*, 2016; Machado *et al*, 2016). Kinetic information, in the form of $k_{cat}$ values, is included as stoichiometric coefficients to convert the metabolic flux in mmol gDWh$^{-1}$ to the required enzyme usage in mmol gDW$^{-1}$. The protein level is included as an upper bound for each enzyme usage; thus, the desired constraint on each flux is respected (Fig 1B).

All enzymes in the model were inferred using the GEM's gene associations and querying SWISS-PROT (Boeckmann *et al*, 2003) and KEGG (Kanehisa & Goto, 2000). Specific formalisms were developed to manage cases such as reversible enzymes, isozymes, promiscuous enzymes, and complexes. Turnover numbers were automatically queried from BRENDA (Schomburg *et al*, 2013) with flexible criteria to manage the high data variability. Enzyme abundances can be set in the model as upper bounds according to experimental values (absolute proteomics). In the absence of proteomic data (or incomplete data), enzyme-specific constraints can be replaced with a total enzyme mass constraint, similar to the FBAwMC approach (Beg *et al*, 2007). For additional details of GECKO, see the Materials and Methods section.

## ecYeast7: an enzyme-constrained model of *Saccharomyces cerevisiae*

### General description of the model

GECKO was applied to the latest version of the consensus genome-scale reconstruction of yeast (Aung *et al*, 2013), Yeast7, which currently consists of 3,493 reactions and 2,220 metabolites. The resulting enzyme-constrained model, hereafter referred to as ecYeast7, has 6,741 reactions and 3,388 metabolites, of which 764 are enzymes and 404 are pseudo-metabolites introduced to manage isozymes (Table 1). Intricate relationships between enzymes and

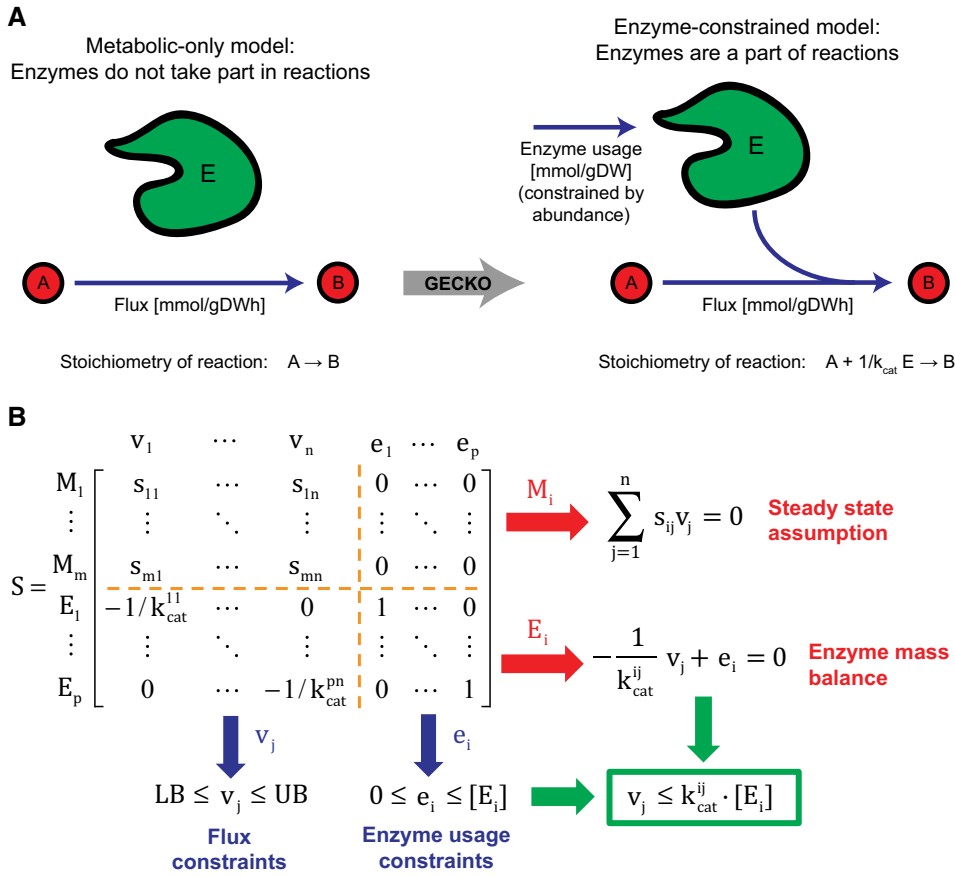

**Figure 1.  Framework for including enzymes as metabolites in a genome-scale model.**

A   GECKO uses a genome-scale model and includes enzymes as part of reactions.

B   Expansion of the stoichiometric matrix. M refers to metabolites, E to enzymes, v to metabolic fluxes, and e to enzyme usage. Note that 4 submatrices appear inside the new stoichiometric matrix: The upper left submatrix is equivalent to the original stoichiometric matrix, the upper right submatrix has only zeros, the lower left submatrix has the kinetic information, and the lower right submatrix is the identity matrix.

reactions are common in the model, with 226 complexes, 373 reactions with isozymes and 315 promiscuous enzymes (Table 1, Appendix Fig S2). Additionally, we can see that enzymes in the cytoplasm, mitochondrion, and endoplasmic reticulum are the most numerous (Fig 2A). Finally, ecYeast7 is fully compatible with the COBRA toolbox (Schellenberger *et al*, 2011) and has similar simulation running times compared to its metabolic counterpart: A standard growth maximization FBA problem of ecYeast7 is solved in $135 \pm 7.3$ ms with the Gurobi LP solver in a Windows PC with a 2.1 GHz Intel i7-4600U two processor, compared to $138 \pm 9.7$ ms in the case of Yeast7.

*Biochemical characteristics of enzymes in the model*

Next, we investigated the biochemical properties, namely $k_{cat}$ values and molecular weights, of the 764 enzymes in ecYeast7. We found that molecular weights spanned 3 orders of magnitude and $k_{cat}$ values spanned 11 orders of magnitude, with median values of 48.2 kDa and 70.9 s$^{-1}$, respectively (Appendix Fig S3). In total, 92.1% of the $k_{cat}$ values were between 0.1 and 10,000 s$^{-1}$, that is, six orders of magnitude (Fig 2B). It should be considered that values in the BRENDA database are typically measured *in vitro* and may thus differ from the *in vivo* values to some extent.

We then compared enzymes among different metabolic functions by classifying them into three different metabolic groups: (i) carbohydrate and energy primary metabolism; (ii) amino acid, fatty acid, and nucleotide primary metabolism; and (iii) intermediate and secondary metabolism (see the Materials and Methods section). Different metabolic groups had different $k_{cat}$ value (Fig 2B) and molecular weight (Fig 2C) distributions. In particular, the carbohydrate and energy primary metabolism enzymes had markedly higher $k_{cat}$ values (median = 120 s$^{-1}$) and lower molecular weights (median = 41.7 kDa) than the other two groups; that is, they were faster in catalysis and smaller. Conversely, the intermediate and secondary metabolism enzymes were slower in catalysis (median $k_{cat}$ value = 45.1 s$^{-1}$) and larger (median molecular weight = 53.6 kDa). Finally, amino acid, fatty acid, and nucleotide primary metabolism enzymes had intermediate $k_{cat}$ values (median = 65.9 s$^{-1}$) and molecular weights (median = 47.1 kDa). This is in agreement with previous observations (Bar-Even *et al*, 2011) that identified central carbon metabolism enzymes as the most efficient in metabolism, most likely due to evolutionary pressure on this part of metabolism to operate with high fluxes.

We also compared biochemical properties across different types of enzymes, that is, complexes, isozymes, and promiscuous

**Table 1.    Descriptors of ecYeast7, the *Saccharomyces cerevisiae* model expanded to account for enzymes.**

| General descriptors of the model | |
| --- | --- |
| Number of reactions | 6,741 |
| Number of metabolites | 3,388 |
| Number of compartments | 14 |
| **Classification of reactions** | |
| Metabolic reactions matched with an enzyme(s) | 3,239 |
| Metabolic reactions not matched with an enzyme | 330 |
| Transport reactions | 1,674 |
| Metabolite exchange reactions | 330 |
| Arm reactions introduced for isozymes | 404 |
| Enzyme usages (treated as reactions) | 764 |
| **Classification of metabolites** | |
| Original metabolites | 2,220 |
| Enzymes | 764 |
| Pseudo-metabolites introduced for isozymes | 404 |
| **Enzyme/reaction relationships** | |
| Complexes | 226 |
| Reactions with isozymes | 373 |
| Promiscuous enzymes | 315 |

enzymes (Appendix Figs S4 and S5). We observed that proteins that belong to complexes are significantly faster (Appendix Fig S4B) and smaller (Appendix Fig S5B) than stand-alone enzymes and that promiscuous enzymes are faster (Appendix Fig S4D) than non-promiscuous enzymes.

*Connectivity of the model*

Connectivity metrics for Yeast7 and ecYeast7 were computed using the metabolite network (Table 2), with and without currency metabolites such as ATP and NADH (see Appendix for more details). Overall, we observed similar values, indicating that the enzyme-constrained model has similar topology to that of the original model. Among the observed differences, the global clustering coefficient and the average betweenness centrality were lower for ecYeast7. This indicates that the enzyme-constrained network is less clustered, which is mainly due to the inclusion of 404 pseudo-metabolites as intermediate steps in reactions with isozymes. Conversely, the average local clustering coefficient is higher than that of Yeast7, which indicates an increase in local clusters. This is consistent with the observed increase in node degree, both in the average (Table 2) and overall distributions (Appendix Fig S6), and is mainly due to the addition of 764 enzymes, which leads to new connections to most metabolites in the network, or several in the case of isozymes.

**Simulating physiological behavior**

First, the model was tested without the input of proteomics data. Only a constraint with the total amount of enzyme (g gDW$^{-1}$) was applied, and the model was allowed to freely allocate the enzymes within this overall constraint. This was performed using a module of GECKO that introduces a pseudo-metabolite that acts as an

enzyme pool (see the Materials and Methods section), creating a total mass constraint similar to the molecular crowding formalism (Beg *et al*, 2007; Adadi *et al*, 2012). With this approach, we tested a series of physiological responses including overflow metabolism, stress response, and consumption of non-typical carbon sources.

*Growth at increasing specific growth rate: simulating the Crabtree effect*

Overflow metabolism occurs in several organisms, including *E. coli* (Van Hoek & Merks, 2012), *S. cerevisiae* (Van Hoek *et al*, 1998), and cancer cells (Vazquez & Oltvai, 2011). In the case of *S. cerevisiae*, it is known as the Crabtree effect; under aerobic conditions at a critical specific growth rate of approximately 0.3 h$^{-1}$ [a value that is strain-dependent (Van Dijken *et al*, 2000)], yeast metabolism switches from purely respiratory to a combination of respiration and fermentation resulting in the production of ethanol, which is less energetically efficient.

There are numerous theories concerning the cause of overflow metabolism (Molenaar *et al*, 2009); recently, it was suggested that protein limitation may be the underlying reason. Faster growth requires more energy and thus higher metabolic fluxes, which requires more enzyme mass. However, the protein pool inside the cell is limited, and respiration enzymes, the main energy generators in the cell, have low specific activity due to their large size. Thus, the cell switches to a more mass efficient protein composition when the specific growth rate surpasses a certain threshold. This composition accounts for pathways that can produce more ATP with the same amount of enzyme mass, even though they have a lower ATP/carbon yield. This has been shown with different approaches in a self-replicator model (Molenaar *et al*, 2009; Berkhout *et al*, 2013), *E. coli* (Vazquez *et al*, 2008; O'Brien *et al*, 2013; Basan *et al*, 2015; Peebo *et al*, 2015), cancer cells (Shlomi *et al*, 2011; Vazquez & Oltvai, 2011), and *S. cerevisiae* (Van Hoek & Merks, 2012; Nilsson & Nielsen, 2016). However, the concept has not been tested for *S. cerevisiae* at the genome-scale using real kinetic values.

Traditional GEMs are unable to show overflow metabolism unless *ad hoc* constraints or objective functions are imposed (Famili *et al*, 2003); therefore, we tested whether ecYeast7 shows this metabolic shift at increasing specific growth rates due to the total enzyme mass constraint. Assuming an average enzyme saturation of 51%, we attained a good fit to experimental data from the literature (Van Hoek *et al*, 1998; Fig 3A), and we observed a region of dual limitation in glucose and enzyme content for dilution rates above 0.3 h$^{-1}$, a region that has been referred to as the Janusian region (O'Brien *et al*, 2013), in which a switch from respiration toward fermentation occurs. As expected, the original Yeast7 model was not able to reproduce this behavior (Appendix Fig S7), and although an enzyme-constrained model with randomly assigned $k_{cat}$ values or molecular weights did occasionally show it (Appendix Fig S8), more than 99.9% of the time the shift was predicted at lower dilution rates. Therefore, we can corroborate that the Crabtree phenotype is an adaptation to the enzyme properties and not a network property (Nilsson & Nielsen, 2016). Additionally, we also see a small Janusian region at high growth rates during anaerobic conditions (Nissen *et al*, 1997; Fig 3D), which shows a slight tradeoff between the ethanol and glycerol production rates.

When investigating the enzyme usage predictions at increasing specific growth rate under aerobic conditions (Fig 3B), we see that

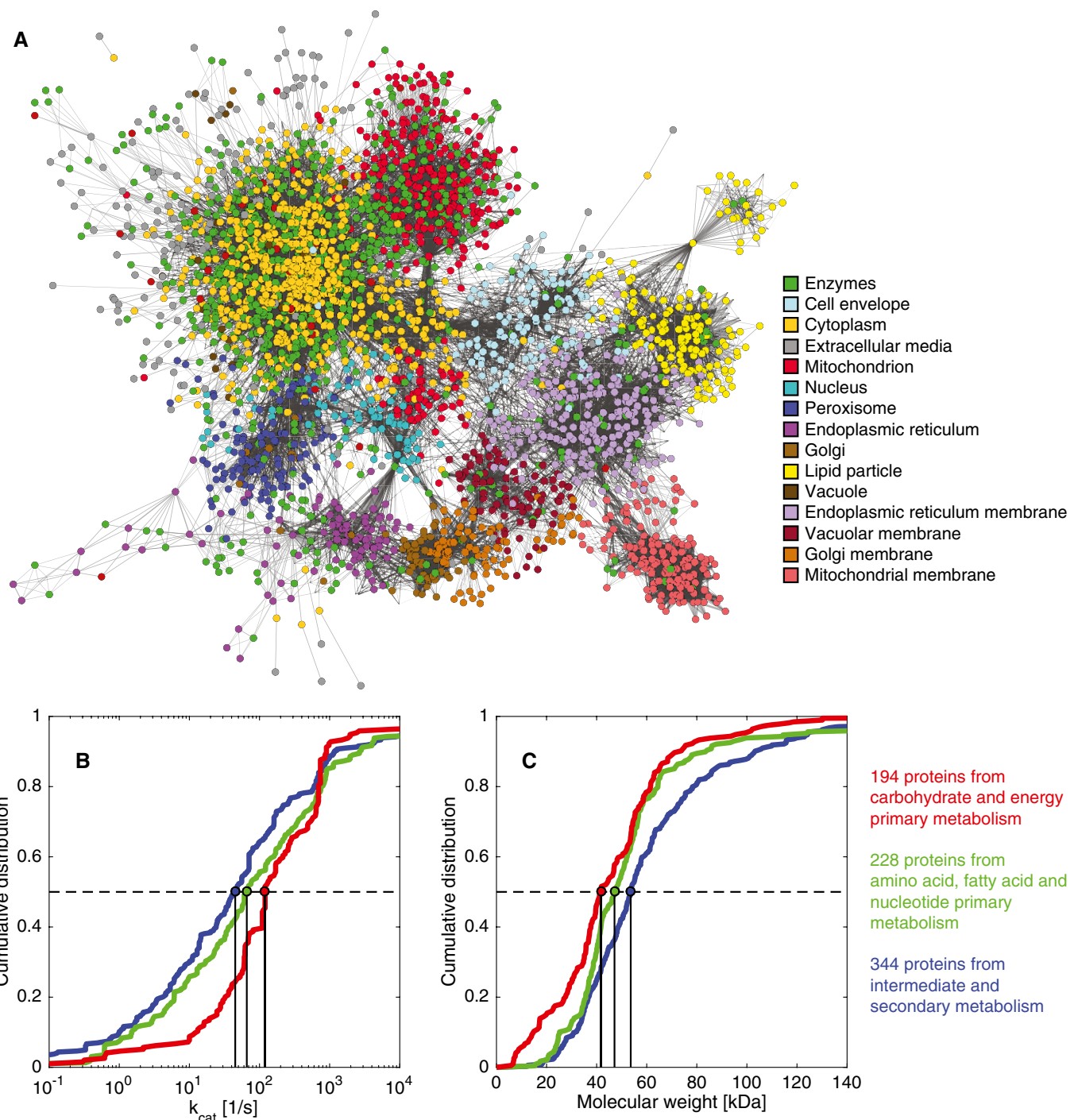

**Figure 2. Visualization of the *Saccharomyces cerevisiae* model expanded to account for enzymes.**

A    Network with metabolites and enzymes, color-coded to show the location of enzymes (green dots) and metabolites (color-coded by compartment).

B, C    Cumulative distributions of (B) $k_{cat}$ values and (C) molecular weights for three different metabolic groups. All distributions were significantly different with $P < 0.05$ using a non-parametric Wilcoxon rank-sum test (Appendix Table S4).

above the critical specific growth rate (0.3 h$^{-1}$), enzymes from the oxidative phosphorylation pathway (the most mass intensive pathway) are progressively replaced by increased abundance of glycolytic enzymes. This supports the view that energy synthesis in

yeast is dominated by two strategies: metabolic efficiency at low specific growth rate and catalytic efficiency at high specific growth rate (Molenaar *et al*, 2009; Nilsson & Nielsen, 2016). Notably, the saturation rather than the concentration could decrease for

**Table 2. Connectivity metrics with and without currency metabolites, computed for both the original metabolic model and the enzyme-constrained model.**

| Metric | Full matrix | | With no currency metabolites | |
|---|---|---|---|---|
| | Yeast7 | ecYeast7 | Yeast7 | ecYeast7 |
| Global clustering coefficient | 0.09 | 0.06 | 0.24 | 0.06 |
| Average local clustering coefficient | 0.58 | 0.63 | 0.41 | 0.56 |
| Average node degree | 13.3 | 13.7 | 8.2 | 9.1 |
| Characteristic path length | 3.4 | 3.4 | 4.6 | 5.1 |
| Diameter | 8 | 9 | 17 | 17 |
| Average path diversity | 9.8 | 10.1 | 12.3 | 41.4 |
| Average betweenness centrality | 1.1E-03 | 7.2E-04 | 2.0E-03 | 1.3E-03 |

oxidative phosphorylation enzymes and increase for glycolytic enzymes (Van Hoek *et al*, 1998, 2000). This distinction is outside the scope of our simulations as we assumed an average saturation among enzymes due to the lack of enzyme-specific saturation data in the literature. Although models that use the overall mass constraint concept have shown to be predictive of protein content to some extent (Nilsson & Nielsen, 2016), good quality genome-scale proteomic datasets of yeast are needed to confirm whether the aforementioned enzyme tradeoffs are related to saturation, concentration or both.

We also investigated whether simulation variability was altered when the enzyme mass constraint was used. We performed flux variability analysis (FVA) (Mahadevan & Schilling, 2003) on the original Yeast7 model and the new ecYeast7 (see Appendix for more details and Appendix Fig S9 for detailed results at different specific growth rates). From the analysis, we see that only a small fraction of the fluxes increased their variability, whereas most of the fluxes either maintained or reduced their variability (Fig 3C). We also see that, up to the critical specific growth rate, between 30 and 40% of the fluxes have a reduced variability when accounting for enzyme constraints, and this increases to over 60% when the enzyme limitation is reached. Overall, we conclude that by accounting for a total enzyme mass constraint in GEMs, we can decrease the intrinsic variability of the model to a large extent.

### Growing under temperature stress

Overflow metabolism in yeast occurs not only at high specific growth rates but also under stress conditions in which high amounts of energy are needed. We simulated stress at high temperatures by increasing the non-growth associated maintenance (NGAM) so that the model would fit experimental data (Lahtvee *et al*, 2016) of a glucose-limited chemostat operated at 0.1 h$^{-1}$ and 38°C, for both Yeast7 and ecYeast7 (Fig 3E). By including enzyme constraints, our model showed secretion of ethanol and a corresponding increased glucose consumption as well as decreased oxygen consumption,

features that are not seen when using the original model (Fig 3E) nor the unmodified NGAM (Appendix Fig S10).

### Maximum growth under different carbon sources

Finally, we tested ecYeast7 to describe aerobic, non-restricted growth on three different media (minimal, with amino acids, or complex) and 12 different carbon sources, and compared the results with literature studies (Tyson & Lord, 1979; Van Dijken *et al*, 2000). Using an average enzyme saturation of 44%, value estimated by fitting the model to growth on glucose, we were able to successfully reproduce the maximum specific growth rate for most conditions (Fig 4B, average relative error of 8% with a *P*-value < 0.001 when comparing the results to models with randomized $k_{cat}$ values). For comparison, we calculated how much growth the purely metabolic model predicted under the same conditions; for this purpose, upper bounds were imposed on all uptake rates based on the results from the enzyme-constrained model, given that unlimited uptake rate leads to unlimited growth in a purely metabolic model. It can be observed that Yeast7 still widely over-predicted growth (Fig 4A, average relative error of 100%), showing that the enzyme constraint can explain the maximum specific growth rate under many different conditions, without any specification of uptake fluxes.

Notably, there was a significantly improved prediction by ecYeast7 for growth on sucrose. Yeast7 predicted a much faster growth on sucrose than growth on glucose (Fig 4A). This is expected because the purely metabolic model will metabolize both glucose and fructose molecules present in sucrose. Interestingly, when simulating with ecYeast7, the specific growth rates on glucose and sucrose are equal (Fig 4B) and in good agreement with those experimentally observed. We examined the flux exchange rates and noticed that none of the fructose content in sucrose is used by ecYeast7, because of the ATP cost associated with fructose uptake. Therefore, to grow optimally with a limited enzyme pool, the cell chooses to utilize the most efficient carbon source, glucose. This is in accordance with the observed *in vivo* behavior of monosaccharide accumulation during sucrose consumption (D'Amore *et al*, 1989) and benchmarks the predictive power of our model. Conversely, trehalose stands out as the worst prediction with the enzyme-constrained model (Fig 4B). The model computes a much higher specific growth rate than that observed experimentally, most likely due to the high hydrated volume of trehalose (Sola-Penna & Meyer-Fernandes, 1998), which reduces protein activity, thereby hindering amino acid/nucleotide uptake, a process that is not captured by our model.

We also compared all of the flux distributions using principal component analysis (PCA). We see that flux simulations from Yeast7 are quite similar among them, with more than 97% of the variability explained with only one component and clustering mainly depending on the specific growth rate (Fig 4C). This indicates that, to a large extent, flux simulations from the purely metabolic model are equivalent but scaled by the specific growth rate. In contrast, simulations of ecYeast7 have more diversity (the first two components comprise ~87% of the variability) and are clustered based on the carbon source; for instance, all three glucose flux distributions are close together (Fig 4D). Furthermore, the flux distributions of non-fermentable carbon sources (ethanol, acetate, glycerol, and galactose) clustered together, pinpointing that similar metabolic

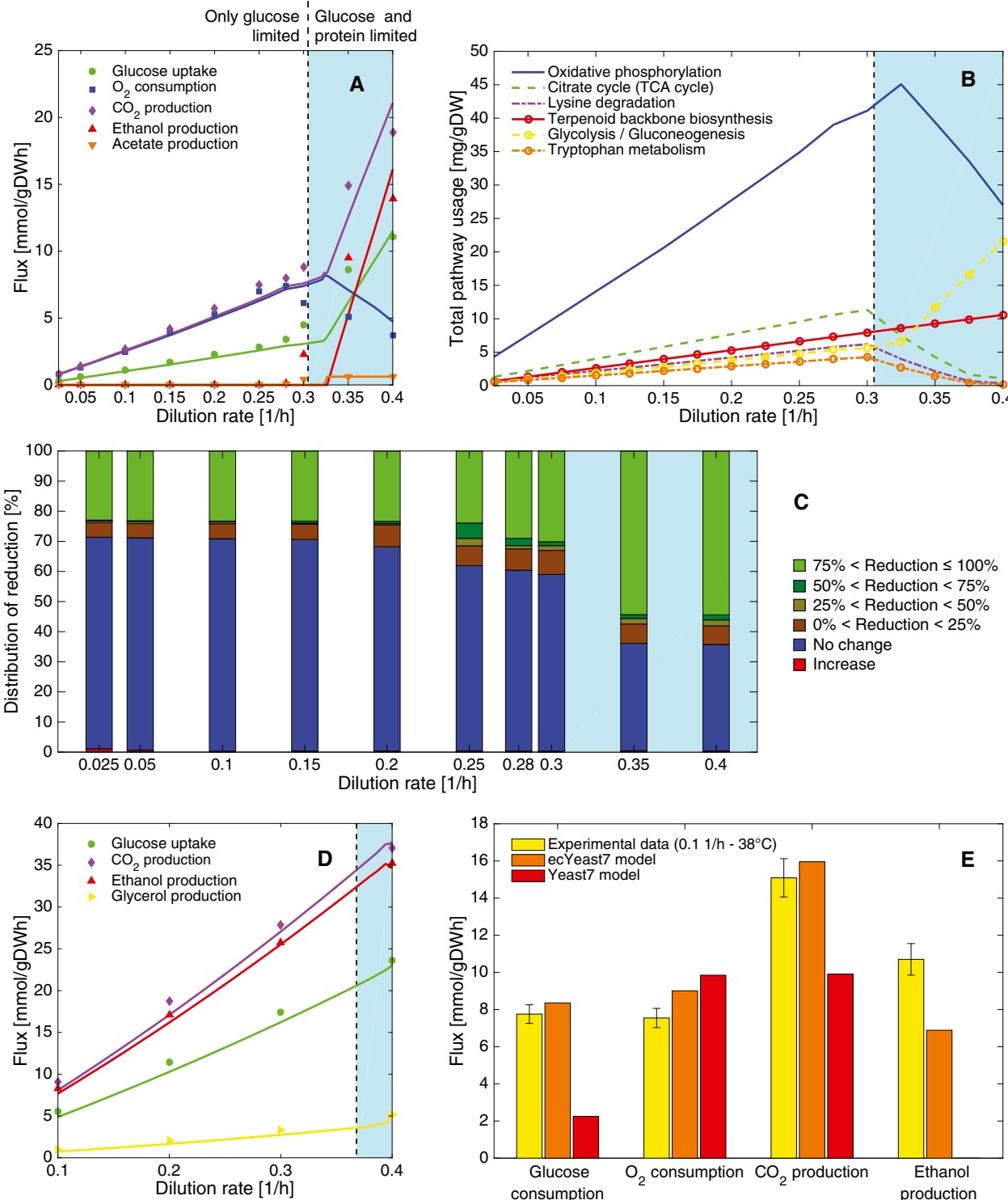

**Figure 3.    The limited enzyme supply defines yeast metabolic strategies at high energy demand in glucose-limited cultures.**

A    Chemostat aerobic data and model simulations at increasing dilution rate.

B    Predicted top 6 pathways used in terms of mass at increasing dilution rate in aerobic conditions.

C    Reduced flux variability compared with purely metabolic GEM simulations at increasing dilution rate in aerobic conditions.

D    Chemostat anaerobic data and model simulations at increasing dilution rate.

E    The model chooses to ferment when energy requirements for non-growth maintenance are high. Experimental data (mean ± SD) were taken from biological triplicates of *S. cerevisiae* grown at high temperature (Lahtvee *et al*, 2016).

Data information: The light blue areas in (A–D) denote the dual-limitation (Janusian) region.

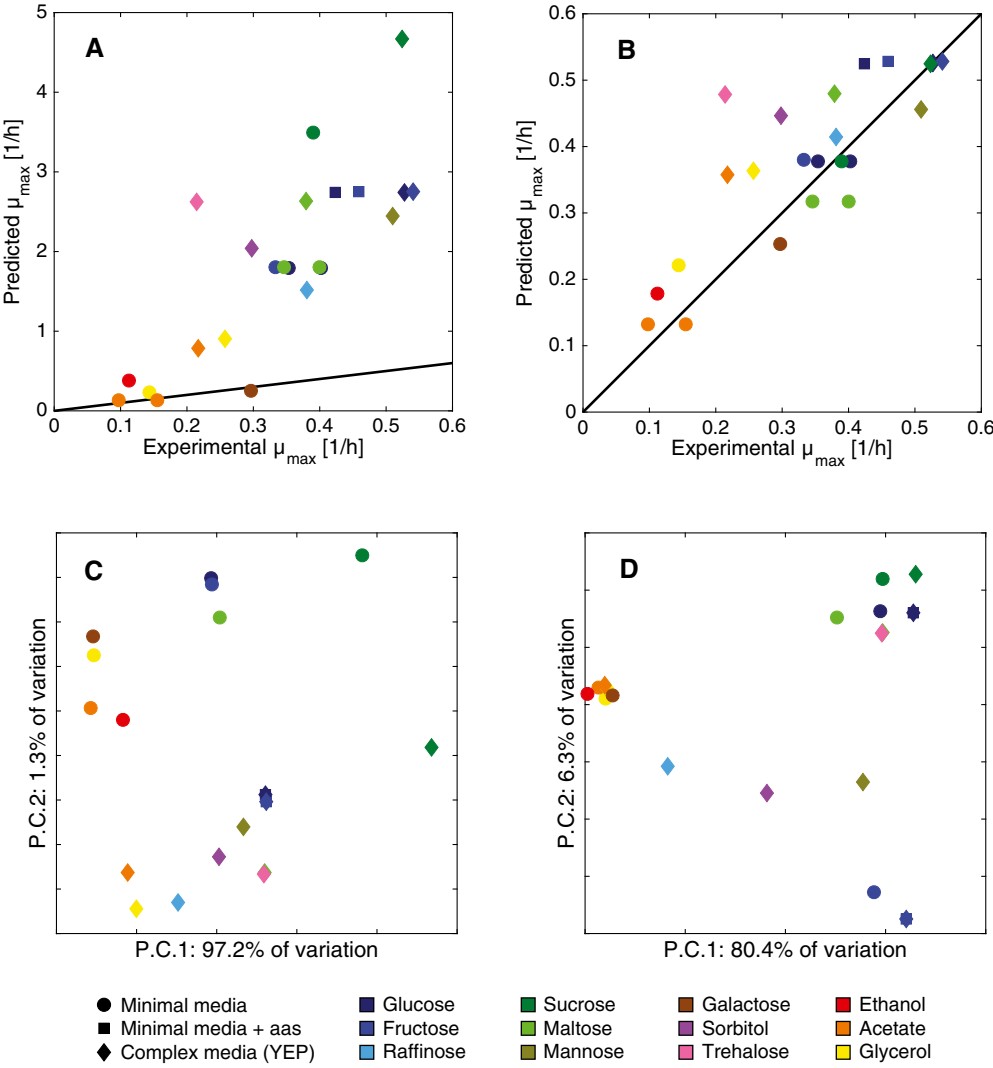

**Figure 4.  The limited enzyme supply defines yeast metabolic strategies in batch growth under different carbon sources.**

A, B   Model simulations of maximum specific growth rate under different aerobic media conditions. (A) Yeast 7 simulations, with an average relative error of 100%. (B) ecYeast7 simulations, with an average relative error of 8%.

C, D   First two components of a PCA for (C) Yeast7 and (D) ecYeast7 flux simulations.

pathways were used in these cases. These observations suggest that the carbon source (main substrate present) plays a more important role in defining the metabolic profile compared to the media type (additional substrates present in small amounts). The developed approach, therefore, gives insight into not only enzyme levels but also metabolic flux distributions.

### Integration of proteomics

GECKO is well suited for direct integration of proteomics; by setting each enzyme abundance, we can run simulations with a more constrained search space. We tested this with an absolute quantitative proteomic dataset of yeast growing at $0.1$ $h^{-1}$ aerobically in a minimal media, limited by glucose (Lahtvee *et al*, 2017). A total of 453 enzymes were directly matched to our model, accounting for

$0.283$ g gDW$^{-1}$ in terms of mass, and the appropriate upper bounds were limited accordingly (including certain flexibility given the variability of the data). For the other 311 enzymes in the model, the overall mass constraint previously mentioned was used, limiting the sum by $0.036$ g gDW$^{-1}$, which represents the remaining mass in the model according to PaxDB (see the Materials and Methods section). This means that out of all enzyme mass in the model, 88.7% directly matches to experimental values and 11.3% does not.

By including proteomics in the enzyme-constrained model, we attained a solution with similar exchange fluxes to those predicted by the purely metabolic model (Appendix Fig S11) and able to predict with similar performance when compared to flux data quantified by $^{13}$C metabolic flux analysis (Jouhten *et al*, 2008; Appendix Table S5 and Fig S12). In order to find the main differences between both flux distributions, we performed random

sampling (Bordel et al, 2010) to both models (details of the random sampling implementation can be found in the Appendix). Out of 4,116 reactions analyzed, 31.7% of them have significantly different ($P < 0.05$) flux values between both models, and only 3.7% of them are both significantly different and have an average difference higher than 0.1 mmol gDWh$^{-1}$. By performing PCA to all flux samples, we observed that 37% of the difference between both model predictions is explained by the first two components (Fig 5A), which are enriched for pyruvate metabolism, fatty acid degradation, and glycerophospholipid metabolism.

We further compared both models by performing FVA (refer to the Appendix for details on the implementation). The analysis yielded that the predictions of ecYeast7 have a significantly lower flux variability than the ones of Yeast7 (Fig 5B, $P = 1.5e-65$ with a non-parametric Wilcoxon rank-sum test). In particular, only 1.5% of the variable reactions in ecYeast7 have complete variability, that is, 1,000 mmol gDWh$^{-1}$, as opposed to the original Yeast7, in which 25.3% of the variable reactions have this freedom. Out of all 4,972 fluxes in the original model (in irreversible format, for a fair comparison), 3,177 had their variability reduced by including enzyme constraints, 1,757 remained the same, and 38 had minor increases (Appendix Fig S13). Among the 3,177 fluxes with reduced variability, the mean reduced variability was 87.7%, and 85.4% had a variability over 90% (Appendix Fig S13B). Overall, by including enzymatic constraints, we significantly decreased the flux variability of simulations while maintaining a physiologically relevant solution.

We also computed the average reduced flux variability by pathway based on the KEGG pathway classification for each enzyme. Out of 60 pathways included in the model, the flux variability decreased in 53 and remained constant in 7. The pathways with decreased flux variability are spread across metabolism (Fig 5C), showing that by using enzyme constraints we refined flux predictions for both efficient pathways carrying high fluxes and inefficient —less utilized—pathways. There is nonetheless a trend to have a higher flux variability reduction among the most utilized pathways, that is, carbohydrate and energy primary metabolism, and a lower flux variability reduction among the least utilized pathways, that is, intermediate and secondary metabolism. This is evidenced by the mean log values of each metabolic group (Fig 5C). Finally, by looking into the pathways with the most reduced flux variability, we see that triglyceride metabolism pathways (glycerolipid and glycerophospholipid metabolism) stand out as the most reduced pathways (Fig 5D). This is due to the fact that Yeast7 has a large amount of triglyceride reactions to represent all possible combinations of fatty acyl triplets; the enzyme limitation thus acts as a capacity constraint in these scenarios, giving more robust results.

### Metabolic engineering applications

Constrained-based modeling techniques such as FBA tend to overestimate biological performance under perturbed conditions, for example, knockout growth and/or production of a specific metabolite of interest (Zhang & Hua, 2016). By accounting for enzyme limitations, the new model should give more realistic predictions under these scenarios. We evaluated this with a case study: a knockout of NDI1, a gene that encodes for the mitochondrial NADH dehydrogenase. Our model was able to capture the shift in the critical specific

growth rate observed *in vivo* (Luttik et al, 2000; Fig 6A, Appendix Fig S14), whereas Yeast7 would miss the effect of the knockout and predict a linear increase in the carbon dioxide production rate over the whole range of dilution rates.

We also compared yield predictions of both succinate and farnesene of ecYeast7 and Yeast7 to experimental data (Raab et al, 2010; Otero et al, 2013; Tippmann et al, 2016). As the production envelopes show during growth on glucose (Fig 6B and C), using our enzyme-constrained model reduced the solution space without leaving out feasible biological solutions. During growth on ethanol, ecYeast7 predicts the same yields for succinate as Yeast7 (Fig 6B), mainly due to the low ethanol uptake rate. It is interesting to note that in the case of succinate production, the identified unfeasible region with the enzyme-constrained model corresponds to mainly high biomass yield, meaning that succinate itself is not a very demanding compound in terms of enzyme utilization. This indicates that classical metabolic engineering approaches are well suited to increase succinate production, such as knocking out genes and coupling production to growth (Patil et al, 2005; Raab et al, 2010). In contrast, in the case of farnesene production, the unfeasible region corresponds to high farnesene yield, meaning that the farnesene production pathway has enzymes with low efficiency. This suggests that a good approach to improve production of farnesene would be to perform protein engineering to improve the activity of enzymes in the corresponding pathway.

As GECKO includes information about enzyme kinetics, it is possible to calculate the so-called flux control coefficients (FCC) (Nilsson & Nielsen, 2016). We analyzed the sensitivity of each enzyme-specific activity on the farnesene specific productivity and hereby calculated the FCCs associated with farnesene production. We found that HMG-CoA reductase and farnesene synthase are the two enzymes with the highest control on farnesene production (Appendix Fig S15) and therefore should be considered as targets for overexpression and/or improvement of enzyme activity. Details on the calculation of FCCs can be found in the Appendix.

## Discussion

We have described GECKO, a simple method for constraining metabolic fluxes with enzymatic data that can be implemented for any GEM. Our method shares elements with previous approaches but stands out as the first method developed for implementing enzyme constraints on a genome-scale model using experimentally measured turnover numbers and enabling the direct integration of absolute proteomic measurements. GECKO is based on the FBAwMC approach (Beg et al, 2007) but extended to limit each individual enzyme, thereby giving a physiologically constrained and thus more feasible solution. On the other hand, as GECKO uses inequalities instead of equalities, it is less constrained than RBA (Goelzer et al, 2015), thus relying less on the quality of the experimental data. Finally, GECKO does not require a detailed description of protein synthesis, and therefore, its implementation to model eukaryal organisms is less demanding compared to the ME-modeling strategy (O'Brien et al, 2013). Furthermore, the resulting enzyme-constrained models have the same structure as any GEM, such that it can be used for any constrained-based analysis method (e.g., FBA, FVA, random sampling), and it can do so in similar computational times compared

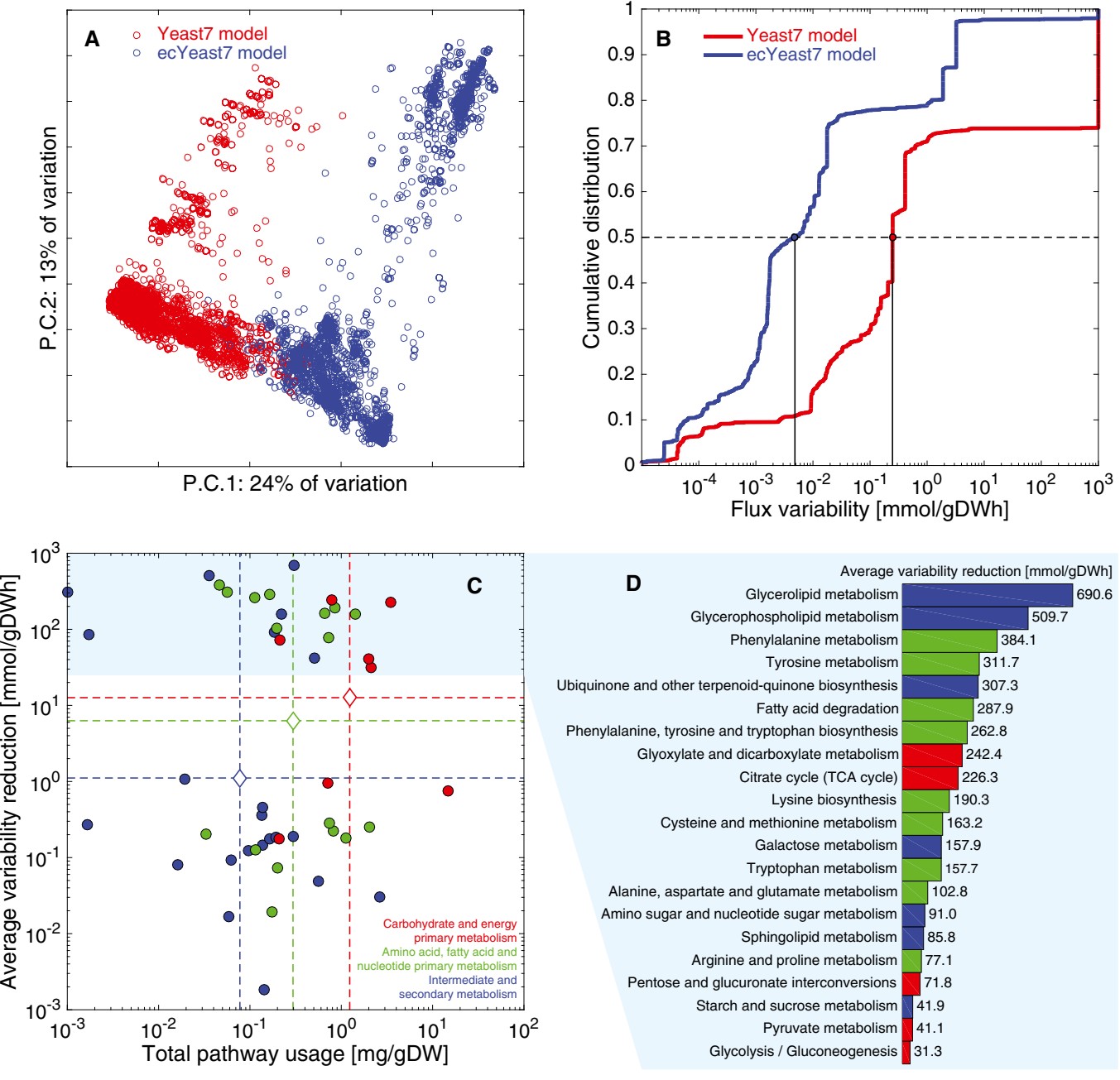

**Figure 5.　Integration of proteomic data into the model.**

A　PCA of fluxes from 10,000 random simulations of Yeast7 and 10,000 random simulations of ecYeast7.

B　Flux variability for all non-zero variable fluxes from Yeast7 (3,286 reactions, 66.1% of all reactions) and ecYeast7 (3,822 reactions, 56.7% of all reactions).

C　Distribution of different pathways in metabolism based on the total usage predicted by ecYeast7 (mg gDW$^{-1}$) and their average flux variability reduction (mmol gDWh$^{-1}$). The discontinuous lines indicate averages of the log values for different metabolic groups. The light blue filled region highlights the pathways with the highest average flux variability reduction shown in (D).

D　Breakdown of the pathways with the highest average flux variability reduction (mmol gDWh$^{-1}$). Colors correspond to the metabolic groups indicated in (C).

to purely metabolic models, further differentiating them from ME models, which require larger computational resources.

By applying GECKO to the latest reconstruction of the yeast consensus metabolic network, we created ecYeast7, an enzyme-constrained model of *S. cerevisiae* in which almost half of the protein mass in yeast is accounted for. By analyzing the

biochemical properties of the accounted enzymes, we showed that energy and carbon metabolism enzymes are significantly faster and smaller than the rest of yeast enzymes. This result agrees with previous observations of $k_{cat}$ values (Bar-Even *et al*, 2011) and is intuitive from an evolutionary context, given that key enzymes should be mass efficient.

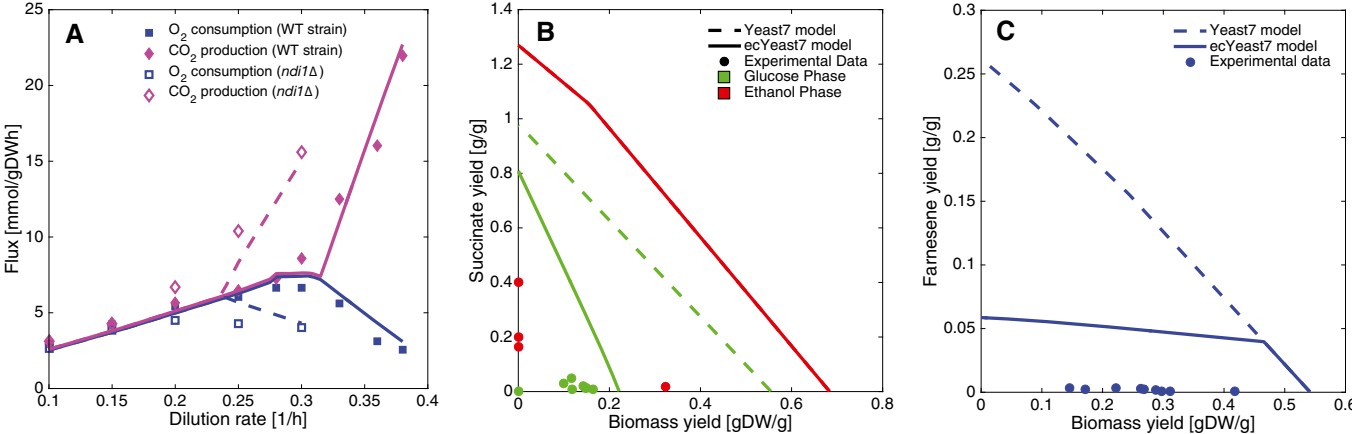

**Figure 6.  Using the enzyme-constrained model for metabolic engineering applications.**

A   The physiological response of knocking out *NDI1* can be reproduced using ecYeast7. Wild-type and knockout experimental data are shown, together with model simulations (continuous and segmented lines, respectively).

B   Succinate yield (grams of succinate per grams of substrate) versus biomass yield (grams of dry weight biomass per grams of substrate) simulations under batch conditions, compared with experimental data. Both glucose and ethanol phases are shown.

C   Farnesene yield (grams of farnesene per grams of glucose) versus biomass yield (grams of dry weight biomass per grams of glucose) simulations under fed-batch conditions, compared with experimental data.

Using ecYeast7 to simulate physiological behavior, we showed that overflow metabolism in yeast may be a consequence of a limited enzyme pool and, ultimately, the energy production capacity of the cell. This identifies enzyme limitation as a major driving force behind the reallocation of enzymatic proteins and the corresponding metabolic fluxes, in agreement with previous studies (Hui *et al*, 2015; Nilsson & Nielsen, 2016). This theory explains not only overflow metabolism but also adaptation under temperature stress and the maximum specific growth rate for several carbon sources under different media conditions. Therefore, our study not only reinforced the hypothesis that efficient proteome reallocation is an important principle in metabolic regulation but also exhibited how simple physico-chemical constraints can be integrated with GEMs to improve their predictive power. Additionally, the enzyme-constrained model proved to be useful for integrating proteomic data and reducing the intrinsic variability of constraint-based simulations. Finally, we showed the potential of the approach in metabolic engineering applications, such as reproducing knockout physiology and improving product yield predictions compared to purely metabolic models. Overall, the GECKO platform is a fundamental tool for improved simulations in quantitative computational biology and is highly useful in basic systems biology for elucidating omics data, and in metabolic engineering for improving the predictive performance of GEMs.

GECKO imposes soft constraints as upper bounds on fluxes, rather than hard constraints for both upper and lower bounds which can over-constrain models. However, one could also consider this as a limitation of the method; by only limiting fluxes with the enzyme's maximum capacity, other processes such as regulation of enzyme activity and under-saturation due to substrate levels are not accounted for. This indicates that biologically unfeasible solutions are still in the solution space; that is, the true solution space is even smaller than the one predicted by ecYeast7. Therefore, a potential next challenge is to include the aforementioned processes in our

approach, which could explain several other metabolic strategies (Noor *et al*, 2016). Metabolite transport/diffusion and protein localization costs could also be accounted for in the future, as recent work has shown that it can improve constraint-based simulations (Liu *et al*, 2014a).

As we have shown, the simulation results of the developed model largely rely on the choice of $k_{cat}$ values (Appendix Fig S8), which could have been incorrectly measured and/or incorrectly annotated. Although it has recently been shown that there is a good correlation between *in vitro* $k_{cat}$ measurements from BRENDA and *in vivo* values inferred from omics data (Davidi *et al*, 2016), the future use of this model for specific purposes such as flux prediction of a given pathway should always involve a preliminary manual curation of the $k_{cat}$ values of the respective pathway. Furthermore, more detailed kinetic data under different conditions, ideally at the genome-scale, will be needed in the future for even more accurate predictions, especially when using this method for organisms with little kinetic information. Special care should be taken to, for instance, distinguish how kinetics vary among different isoforms of the same gene, in the case of eukaryal organisms that exhibit splice variants. It is worthy to mention here that no splice variants are reported for any of the genes in the Yeast7 model.

We envision many possible future uses of the developed ecYeast7 model. For instance, by overlaying proteomic data, one could easily compute $v_{max}$ values, the ultimate constraints on reactions, which otherwise have to be measured individually with kinetic assays. Additionally, by comparing enzyme measurements to their usage in the model, we could compute enzyme usage percentages, which can be interpreted as a new layer of information connecting proteomics and fluxomics, and can be studied to find usage trends among different experimental conditions. Moreover, by having several experimental conditions, we could find enzymes that are highly used among all conditions, which could be a sign of

transcriptional regulation. Finally, the developed model could also be used for testing the effects of varying specific enzyme levels on the production of a metabolite of interest, modifications that could be tested *in vivo* with recently developed techniques for fine-tuning transcription (Farzadfard *et al*, 2013).

# Materials and Methods

## Additional details of GECKO

GECKO uses the flux balance analysis (FBA) approach (Orth *et al*, 2010). In FBA, the stoichiometry of the cell's metabolism is represented in a stoichiometric matrix. The columns of this matrix indicate the stoichiometry of reactions, and the rows indicate mass balances for each metabolite. By assuming pseudo-steady-state conditions (i.e., no accumulation), imposing constraints on fluxes and assuming a cellular objective function, an optimal solution for all metabolic fluxes can be found. In this study, we further constrained the solution subspace by limiting fluxes with enzyme levels. For any enzyme $E_i$ that catalyzes a reaction $R_j$, it holds true that:

$$v_j \leq k_{cat}^{ij} \cdot [E_i] \tag{1}$$

where $v_j$ is the metabolic flux of $R_j$ (mmol gDWh$^{-1}$), $[E_i]$ is the intracellular concentration of $E_i$, and $k_{cat}^{ij}$ is the turnover number of $E_i$ catalyzing $R_j$. With GECKO, we impose this constraint on each reaction in the model (Fig 1). The procedure also accounts for different relationships between enzymes and reactions, as described in the following:

(i)     If the reaction is *reversible*, two reactions are defined, one in the forward direction and one in the backward direction, both utilizing the same enzyme but possibly with specific $k_{cat}$ values, depending on the enzyme's substrate affinity.

(ii)    In the case of a reaction having *isozymes*, one reaction for each enzyme is specified in the model, with different $k_{cat}$ values based on affinity, when available. Additionally, to keep the same original upper bound in the reaction's flux, an "arm reaction" is introduced to constrain the overall flux, creating a pseudo-metabolite that acts as an intermediate between the substrates and the products, as previously introduced (Zhang *et al*, 2015).

(iii)   If an enzyme is *promiscuous*, then the same enzyme will be used by all the respective reactions, possibly with different $k_{cat}$ values (because of different substrates). This implies that there will be more than one non-zero coefficient in some rows of the lower left submatrix of the stoichiometric matrix (Fig 1B). Additionally, because only one enzyme utilization constraint is defined, the reactions will share the amount of enzyme available.

(iv)    Finally, in the case of a *complex*, the reaction will utilize all of the subunits belonging to it. This implies that there will be more than one non-zero coefficient in some columns of the lower left submatrix of the stoichiometric matrix (Fig 1B). Additionally, each subunit's stoichiometric coefficient in the reaction will be multiplied by the corresponding subunit's stoichiometry in the complex.

With these formalisms, complicated constraints (Adadi *et al*, 2012) are circumvented, and we can directly overlay proteomic data in the form of an abundance vector on top of the enzymes' usage in the model. It should be noted that enzymes are not consumed in our approach, but rather occupied; given that we are operating under the steady-state assumption, for a fraction of a second, there is a limited amount of enzyme occupied by its substrates to catalyze the corresponding flux. Therefore, by imposing a mass constraint on the enzyme level, our framework prevents reactions from having higher fluxes than that allowed by the enzyme concentrations.

The developed framework is explained in further detail in the Appendix, where an example for a toy model is also supplied (Appendix Fig S1). GECKO is implemented in MATLAB, with a small section implemented in Python (for querying $k_{cat}$ values from BRENDA). Both GECKO and ecYeast7 are compatible with the COBRA toolbox (Schellenberger *et al*, 2011) or any constraint-based approach.

## Criteria for obtaining kinetic data

All $k_{cat}$ values were automatically retrieved from the BRENDA database (Schomburg *et al*, 2013). Several criteria were implemented to properly match each enzyme/reaction pair in the model to its corresponding $k_{cat}$ value (Appendix Table S3). In case of missing data, certain flexibility was introduced by matching the $k_{cat}$ value to other substrates, organisms, or even introducing wild cards in the EC number. In case that more than one value was available, the maximum $k_{cat}$ value (which corresponds to the fastest working enzyme) was chosen to avoid over-constraining the model (Nilsson & Nielsen, 2016). Finally, the $k_{cat}$ values of the main metabolic pathways were manually curated with previous data (Nilsson & Nielsen, 2016) and further literature search. Additional information regarding $k_{cat}$ retrieval and matching can be found in the Appendix.

## Corrections to the yeast genome-scale model

Several corrections were made to the genome-scale metabolic model of yeast (Aung *et al*, 2013) before its use, such as correcting the glucan coefficients in the biomass pseudo-reaction (Appendix Table S1), accounting for an extracellular membrane potential and correcting coefficients in the oxidative phosphorylation pathway. Energy maintenance was also corrected to improve predictive performance: The growth associated maintenance (GAM) without accounting for polymerization costs was refitted to 31 mmol gDWh$^{-1}$ under aerobic conditions based on experimental data (Van Hoek *et al*, 1998), and the NGAM was set at 0.7 mmol gDWh$^{-1}$ (Nilsson & Nielsen, 2016). See the Appendix for more details.

For simulating anaerobic conditions, the oxygen uptake in the model was blocked and fatty acids and sterols were supplied to the media. Additionally, heme was removed from the biomass pseudo-reaction (given that its synthesis requires oxygen and that it is not needed for anaerobic growth), some reactions' reversibilities were modified for proper glycerol production, and the growth associated maintenance was set to 16 mmol gDWh$^{-1}$ without accounting for polymerization costs. Refer to the Appendix for additional details.

## Managing the lack of proteomic data

In the absence of proteomic data, we limited the total amount of enzyme instead of each enzyme separately. To achieve this, we introduced an additional pseudo-metabolite that represents an aggregated pool of all enzymes present in the model. This pseudo-metabolite's usage has an upper bound equal to the total protein content $P_{total}$ (g gDW$^{-1}$) multiplied by a fraction $f$, which represents the mass fraction of enzymes that are accounted for in the model according to PaxDB (Wang *et al*, 2012) [equal to 0.4461 g (protein)/g (total cellular protein) in the case of Yeast 7.6], and a parameter $\sigma$ representing the average *in vivo* saturation of all enzymes. Additionally, we included reactions that draw mass from this pool toward each enzyme. Hence, a mass balance for the enzyme pool yields:

$$\sum_{i}^{P} MW_i\, e_i \le \sigma \cdot f \cdot P_{total} \qquad (2)$$

This is similar to a previous approach that accounts for enzyme limitation (Adadi *et al*, 2012). When using this formalism, additional considerations were made, such as adjusting the amino acid and carbohydrate composition of the biomass according to experimental data (Nissen *et al*, 1997; Van Hoek *et al*, 1998), and rescaling the polymerization costs in the growth associated maintenance (GAM) (details are in the Appendix).

## Model analysis

Cytoscape (Cline *et al*, 2007) was used to display the metabolite network of the model; metabolites were defined as nodes and an edge between two nodes was created if both metabolites were present in the same reaction. The edge thicknesses were set according to the number of shared reactions. Pseudo-metabolites created because of isozymes were not colored. The prefuse force directed layout algorithm was used for laying out the network.

The enzymes in the model were classified depending on the pathways they belong in the KEGG database (Kanehisa & Goto, 2000). KEGG pathways were classified into one of three metabolic groups according to a previous classification (Bar-Even *et al*, 2011; Appendix Table S2). The intermediate and secondary metabolism groups were considered just as one metabolic group given how few secondary metabolism enzymes there were in the model.

## Simulation details

### Chemostat growth simulations

Chemostat growth was simulated by fixing the specific growth rate to a given dilution rate, limiting the total enzyme mass to experimental measurements and minimizing the substrate's uptake rate. Subsequently, the substrate's uptake rate was fixed at the obtained value and the enzyme usage was minimized, similarly to the parsimonious FBA procedure (Lewis *et al*, 2010) but with the enzyme content. An average enzyme saturation of 46% was used in the case of CEN-PK113-7D simulations (value fitted to aerobic chemostat data [Luttik *et al*, 2000]) and 51% for other strains [value fitted to aerobic chemostat data of the DS28911 strain (Van Hoek *et al*, 1998)]. Finally, some exchange reactions such as pyruvate and

acetate exchange were limited to experimental values, and the reversibility of some NADPH-associated reactions was corrected based on previous work (Pereira *et al*, 2016); see the Appendix for more details.

### Batch growth simulations

For simulation of growth in batch cultures, that is, unlimited substrate availability, no additional constraints aside from the protein limitation were needed, because during the exponential phase substrate is available in excess. All experimental data used were obtained from shake flask cultures from the literature (Tyson & Lord, 1979; Van Dijken *et al*, 2000), and oxygen-excess conditions were assumed. Therefore, unless otherwise stated, the procedure was to remove any constraint on substrate uptake, change the media accordingly (either minimal, with amino acids or complex), limit the total enzyme mass, and parsimoniously maximize biomass. An average enzyme saturation of 44% was assumed for all batch conditions (fitted parameter to growth on minimal media with glucose as carbon source). An upper bound of 2 mmol gDWh$^{-1}$ for the amino acids and/or nucleotides uptake rates in non-minimal conditions was imposed. See the Appendix for more details.

### Proteomic integration

Absolute proteomic measurements were obtained from a recent study (Lahtvee *et al*, 2017). Briefly, *S. cerevisiae,* strain CEN.PK113-7D, was grown aerobically in glucose-limited minimal media conditions at 0.1 h$^{-1}$, in triplicate. The biomass for proteome analyses was collected within 70 s by centrifuging 2 ml of culture broth at 4°C, discarding the supernatant and snap-freezing the pellet in liquid nitrogen. A lysine auxotrophic strain was used to create fully labeled biomass by feeding labeled heavy $^{15}$N, $^{13}$C-lysine (Cambridge Isotope Laboratories Inc., Tewksbury, MA, USA), which was absolutely quantified against the UPS2 protein mix and used as an internal standard in the proteome analysis.

The samples were digested with 1:50 LysC overnight at room temperature. Injected peptides (2 μg) were separated on an Ultimate 3000 RSLCnano system (Dionex, Sunnyvale, CA, USA) using a C18 cartridge trap column in a backflush configuration and an in-house packed (3 μm C18 particles, Dr. Maisch) analytical 50 cm × 75 μm emitter column (New Objective). Following a LC separation, the peptides were eluted to a Q Exactive (Thermo Fisher Scientific) tandem mass spectrometer operating with a top-10 strategy and a cycle time of 0.9 s. The raw data were identified and quantified with the MaxQuant 1.4.0.8 software package. The heavy spike-in standard was quantified with the iBAQ option enabled using log fit (Schwanhäusser *et al*, 2011). The peptide-spectrum match and the protein false discovery rate (FDR) were maintained below 1% using a target-decoy approach. All proteomic data and further details on sample preparation and measurement are available online via the PRIDE repository (Vizcaíno *et al*, 2013), dataset identifier PXD005041. For this study, values were converted to units of mmol gDW$^{-1}$, and each enzyme usage in the model was bounded by:

$$e_i \le \mu_i + \sigma_i \qquad (3)$$

where $\mu_i$ is the mean and $\sigma_i$ the standard deviation of $E_i$ concentration among the triplicates. Proteins not detected in 2 or more of the triplicates were not considered in the analysis. The oxidative

phosphorylation complexes measurements were corrected to be proportional to the average relative abundance of all subunits, given that some subunits were detected in very low amounts.

The measured total protein content was 0.448 g gDW$^{-1}$ (Lahtvee *et al*, 2017), from which 0.283 g gDW$^{-1}$ was matched in total to the model with equation (3). Therefore, equation (2) was used for the unmeasured enzymes as a constraint with 0.448–0.283 = 0.165 g gDW$^{-1}$ as the amount of enzyme, corrected with $f = 0.2154$ g g$^{-1}$ (which represents the abundance of the 311 enzymes in the model from all proteins not matched to the model) and $\sigma = 46\%$. See the Appendix for more information.

### Knockout simulations

The *NDI1* (YML120C) knockout in ecYeast7 was simulated by blocking the usage of the associated enzyme (P32340). Additionally, both cytosolic NADH dehydrogenases *NDE1* (YMR145C–P40215) and *NDE2* (YDL085W–Q07500) were blocked to simulate the limited capacity of the ethanol-acetaldehyde shuttle *in vivo* (Luttik *et al*, 2000), which is not described in BRENDA. Finally, an average enzyme saturation of 46% was assumed, given that the strain CEN.PK113-7D was used in the experimental data source study (Luttik *et al*, 2000).

### Yield simulations

For the succinate case study, experimental yields from different strains were gathered from the literature (Raab *et al*, 2010; Otero *et al*, 2013) for growth on glucose and ethanol during batch cultivations. Glucose and ethanol consumption rates of 8.459 and 0.806 mmol gDWh$^{-1}$, respectively, were assumed [calculated from the wild-type strain growth (Raab *et al*, 2010)]. For the farnesene case, the experimental yields from different strains under fed-batch growth (Tippmann *et al*, 2016) were used as validation, and a glucose consumption rate of 1.228 mmol gDWh$^{-1}$ was assumed (highest value reported in the study). Additionally, two modifications were performed in ecYeast7 for the case of farnesene: the addition of the farnesene synthase from *Malus domestica* [Tippmann *et al*, 2016) ($k_{cat} = 0.0553$ s$^{-1}$ according to the literature (Green *et al*, 2009)], and the truncation of HMG-CoA reductase to avoid cleavage to the mitochondrial membrane (Polakowski *et al*, 1998), which reduces the enzyme weight to 46% of the original weight.

Both case studies were performed by fixing the specific glucose uptake rate and maximizing the biomass yield, fixing the specific growth rate to a suboptimal value and maximizing the corresponding compound production rate, and then fixing the compound production rate and minimizing enzyme usage. An average enzyme saturation of 44% was assumed (value for batch conditions).

### Data and software availability

The data used in this study and the developed method and model are all available online:

- Absolute proteomic data: PRIDE PXD005041, "Absolute quantification of yeast proteome" (http://www.ebi.ac.uk/pride/archive/projects/PXD005041; only reference conditions were used)
- GECKO method: GitHub (https://github.com/SysBioChalmers/GECKO/releases/tag/v1.0)
- ecYeast7 model (both constrained and unconstrained; each as .mat, .sbml and .txt files): GitHub (https://github.com/SysBioChalmers/GECKO/tree/v1.0/Models)

**Expanded View** for this article is available online.

### Acknowledgements

The authors would like to thank Michael Gossing, Sunjae Lee, Johan Björkeroth, Amir Feizi, and Henning Redestig for valuable input. This project has received funding from the European Union's Horizon 2020 research and innovation program under grant agreements No 686070 and 720824, the Novo Nordisk Foundation, the Knut and Alice Wallenberg Foundation, and the US Department of Energy, Office of Science, Office of Biological and Environmental Research, Genomic Science program (DE-SC0008744). B.J.S. gratefully acknowledges financial support from CONICYT (grant #6222/2014).

### Author contributions

JN, BJS, and CZ conceived the project. BJS and CZ developed the computational method. P-JL performed the protein quantifications. BJS performed the computational simulations and analysis of the results. AN, CZ, and EJK contributed to analysis of the results. BJS wrote the draft of the paper. All authors read, edited, and approved the final manuscript.

### Conflict of interest

The authors declare that they have no conflict of interest.

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
