## [Review Process File · Molecular Systems Biology]

Improving the phenotype predictions of a yeast genome-scale metabolic model by incorporating enzymatic constraints

Benjamín J. Sánchez, Cheng Zhang, Avlant Nilsson, Petri-Jaan Lahtvee, Eduard J. Kerkhoven & Jens Nielsen

Corresponding author: Jens Nielsen, Chalmers University of Technology

Review timeline:	Submission date:	26 October 2016
	Editorial Decision:	24 January 2017
	Revision received:	13 April 2017
	Editorial Decision:	14 June 2017
	Revision received:	14 June 2017
	Accepted:	19 June 2017

Editor: Thomas Lemberger

Transaction Report:

1st Editorial Decision

24 January 2017

Thank you again for submitting your work to Molecular Systems Biology. We have now finally heard back from the referees who agreed to evaluate your manuscript. As you will see from the reports below, the referees find the topic of your study of potential interest. They raise, however, several issues, which should be convincingly addressed in a revision of the work.

Without repeating all the points raised by the reviewers, the major issue refers to the need of a deeper analysis and the reviewers provide some suggestions in this regard.

We are aware that referee #2 suggests to add some additional discussion in supplementary information. We would however be reluctant to extend or add discussion elements outside of the main paper. If the new discussion points are important, please include it in the main text, otherwise, if the issues are peripheral, we would agree that such points can be omitted.

With regard to the ecYeast7 model resulting from the GECKO analysis, we would like to send the results for validation/curation such that it can be verified during the revision process. The model is however now only provides in binary Matlab format. We would thus kindly ask you to send us as soon as possible the model in SBML, if possible.

When you resubmit your manuscript, please download our CHECKLIST (<http://embopress.org/sites/default/files/Resources/EP_Author_Checklist_Master.xlsx>) and include the completed form in your submission. *Please note* that the Author Checklist will be published alongside the paper as part of the transparent process <<http://msb.embopress.org/authorguide#transparentprocess>>.

If you feel you can satisfactorily deal with these points and those listed by the referees, you may wish to submit a revised version of your manuscript. Please attach a covering letter giving details of the way in which you have handled each of the points raised by the referees. A revised manuscript will be once again subject to review and you probably understand that we can give you no guarantee at this stage that the eventual outcome will be favorable.

REFEREE REPORTS

Reviewer #1:

Here the authors propose a set of new constraints for use in flux balance analysis of genome-scale metabolic models. These constraints account for the flux capacity of enzymatic reactions. They apply these constraints with a model of yeast, then demonstrate how a model with these constraints is capable of generating predictions that a standard model cannot produce. Interestingly, the constraints are all encoded as additional reactions and stoichiometric coefficients in the model, meaning the new model is completely compatible with existing FBA software.

The idea of adding constraints to FBA to account for enzyme capacity and overall enzyme availability has been thoroughly explored in previous work, and the authors did an excellent job of reviewing this work in their introduction. The authors also did a good job of contrasting previous methods with their own technique. Despite all of these existing methods, it appears as though what the authors are presenting is a novel and significant advance. The authors also present a compelling set of demonstrations for their model, showing how the model with their enhanced constraints can capture biology that the existing model cannot.

Overall, this is an extremely well-written manuscript, with a very clearly and concisely described approach, and it represents a significant advance to the art.

I have only two significant comments, and one minor comment.

Significant comments:

1.) The authors have done a great job of showing a number of case-studies where their new technique captures biology that standard FBA approaches cannot. However, it would be useful if the authors also compared the enhanced predictive capabilities of their algorithm with other existing similar techniques (e.g. ME modeling, FBAwMC, RBA). Although it is certainly out of scope to actually apply the competing methods to each of the described case studies, it would be useful if the authors could comment on whether or not the existing competing methods would be expected to perform worse, the same, or better than GECKO. Without this discussion, it's difficult to fully evaluate GECKO vs these competing methods.

2.) Models commonly map genes to metabolic reactions, while the new constraints introduced by the authors relate to proteins. In prokaryotes, there is typically a 1-1 relationship between genes and proteins, but in eukaryotes, this is not the case. Due to splice variants, an individual gene may map to several different proteins, likely with different cat values. Can the authors comment on this? Is their model mapped to protein IDs or genes? Is it expected that this might impact results? How might this impact the proteomics-based analysis, if at all?

Minor comments:

1.) It could be I missed it somewhere in the manuscript, but it appears as though the authors never indicate how many of model reactions they were able to find kcat values for in BRENDA. Did kcat values exist for all reactions? If not, what kcat value was used for reactions where no measured value could be found? How many kcat values were not exact matches but exploited the "flexible" matching to brenda mentioned in the methods? Are the measured kcat values all collected for similar/identical enzymes?

Reviewer #2:

Overall, the formulation presented in this is elegant and represents a very good contribution to the field. Moreover, the model provided is also a relevant tool for the community.

However, the full formulation of the method with individual enzyme levels has not been sufficiently exploited and the result shown of decrease flux variability seems somewhat obvious. Although the impact of decreasing flux variability in metabolic engineering design has been explored, I would expect a wider variety of analysis here.

The results shown for the more general constraint of total enzyme mass are somewhat more consistent and span a variety of effects. However, novelty in the approach here is somewhat more limited compared with existing methods, models and concepts. Thus, I believe the authors need to expand the results section for the full formulation of the method such that this paper can bring the expected level of contribution for MSB.

Moreover, in section 2.4 it is essential to know if the experiments with Yeast 6 have been performed with pFBA or FBA. Given the high variability found, I suspect they were performed with FBA and, given the formulation of GECKO, inspired in pFBA, I believe this is not a fair comparison

Some minor concerns:

Introduction

Sentence "However, when considering the production of a metabolite of interest, these models typically make the assumption that the uptake rate of the carbon source (e.g., glucose) limits production. This may be an oversimplification because experimental yields are usually considerably lower than the maximum theoretical yields⁵"

Although the first sentence is correct, it does not imply the second observation. Usually in Metabolic Engineering design projects the maximum theoretical yield is not assumed.

In terms of the present formulation, the authors need to refer to the paper Machado et al 2016 PLOS Comput Biol given the obvious similarities (although overall the GECKO method brings a sufficient level of innovation)

Results

Overall, sections 222 and 223 are of moderate interest but I feel some observations would need to be discussed. As this would probably expand this section, it might make sense to put part of it as supplementary material. As an example: Why proteins in complexes and promiscuous are faster and smaller (for complexes)? Also, the network properties also require further discussion

Regarding data in Figure 3E - I miss a comparison with predictions made with nominal maintenance

Growth rates (233)

For comparison purposes, the authors either use their method with random k_{cat} values or use normal FBA simulations using the uptake rates given by the GECKO simulation. How would the original model behave in FBA in defined medium if all input fluxes are experimentally given (as this is the standard simulation method)? Although for complex medium this might be difficult to get, it is usually obtained in experiments using defined media at least for the carbon source uptake rate. A related question: how do the uptake rates obtained with the new model / formulation compare with the ones observed experimentally?

Also in this section, flux values could have been compared with experimental ¹³C data, for example for one fermentable and one non-fermentable carbon source

Section 2.5

"Given their unconstrained nature, GEMs tend to overestimate biological performance such as knockout growth and/or production of a specific metabolite of interest."

I believe there is a confusion with model and simulation. In this case, other simulation tools (non FBA) such as MOMA or ROOM provide less overestimates for biomass performance.

Reviewer #1:

Here the authors propose a set of new constraints for use in flux balance analysis of genome-scale metabolic models. These constraints account for the flux capacity of enzymatic reactions. They apply these constraints with a model of yeast, then demonstrate how a model with these constraints is capable of generating predictions that a standard model cannot produce. Interestingly, the constraints are all encoded as additional reactions and stoichiometric coefficients in the model, meaning the new model is completely compatible with existing FBA software.

The idea of adding constraints to FBA to account for enzyme capacity and overall enzyme availability has been thoroughly explored in previous work, and the authors did an excellent job of reviewing this work in their introduction. The authors also did a good job of contrasting previous methods with their own technique. Despite all of these existing methods, it appears as though what the authors are presenting is a novel and significant advance. The authors also present a compelling set of demonstrations for their model, showing how the model with their enhanced constraints can capture biology that the existing model cannot.

Overall, this is an extremely well-written manuscript, with a very clearly and concisely described approach, and it represents a significant advance to the art.

We thank the reviewer for the kind words.

I have only two significant comments, and one minor comment.

Significant comments:

1.) The authors have done a great job of showing a number of case-studies where their new technique captures biology that standard FBA approaches cannot. However, it would be useful if the authors also compared the enhanced predictive capabilities of their algorithm with other existing similar techniques (e.g. ME modeling, FBAwMC, RBA). Although it is certainly out of scope to actually apply the competing methods to each of the described case studies, it would be useful if the authors could comment on whether or not the existing competing methods would be expected to perform worse, the same, or better than GECKO. Without this discussion, it's difficult to fully evaluate GECKO vs these competing methods.

We have followed the reviewer's advice and included in the discussion a comparison with the mentioned 3 approaches (section 3, 1st paragraph), as such:

“GECKO is based on the FBAwMC approach but extended to limit each individual enzyme, thereby giving a physiologically constrained and thus more feasible solution. On the other hand, as GECKO uses inequalities instead of equalities, it is less constrained than RBA, thus relying less on the quality of the experimental data. Finally, GECKO does not require a detailed description of protein synthesis, and therefore its implementation to model eukaryal organisms is less demanding compared to the ME-modeling strategy. Furthermore, the resulting models have the same structure as any GEM, such that it can be used for any constrained-based analysis method (e.g. FBA, FVA, random sampling, etc.); and it can do so in similar computational times compared to purely metabolic models, further differentiating them from ME-models, which require larger computational resources.”

2.) Models commonly map genes to metabolic reactions, while the new constraints introduced by the authors relate to proteins. In prokaryotes, there is typically a 1-1 relationship between genes and proteins, but in eukaryotes, this is not the case. Due to splice variants, an individual gene may map to several different proteins, likely with different cat values. Can the authors comment on this? Is their model mapped to protein IDs or genes? Is it expected that this might impact results? How might this impact the proteomics-based analysis, if at all?

We agree that splice variants could eventually impact results, however in yeast the amount of splice variants is very low. In particular, no splice variants are reported in Swissprot for any of the genes in the Yeast7 model, therefore our model uses always the correct match gene-protein. The reviewer's observation is nonetheless a very important consideration if this method is to be implemented in other organisms with a higher frequency of splice variants and we therefore added a comment about this in the discussion (5th paragraph), as such:

“[...]Special care should be taken to, for instance, distinguish how kinetics vary among different isoforms of the same gene, in the case of eukaryal organisms that exhibit splice variants. It is worthy to mention here that no splice variants are reported for any of the genes in the Yeast7 model.”

Minor comments:

1.) It could be I missed it somewhere in the manuscript, but it appears as though the authors never indicate how many of model reactions they were able to find kcat values for in BRENDA. Did kcat values exist for all reactions? If not, what kcat value was used for reactions where no measured value could be found? How many kcat values were not exact matches but exploited the "flexible" matching to brenda mentioned in the methods? Are the measured kcat values all collected for similar/identical enzymes?

Overall our method extracted 3249 values from BRENDA, from which more than 90% come from using at most 1 wild card, and more than 50% are values from *S. cerevisiae*. Further details can be found in supplementary table S3 (section 4.1 in the supplementary material).

Reviewer #2:

Overall, the formulation presented in this is elegant and represents a very good contribution to the field. Moreover, the model provided is also a relevant tool for the community.

However, the full formulation of the method with individual enzyme levels has not been sufficiently exploited and the result shown of decrease flux variability seems somewhat obvious. Although the impact of decreasing flux variability in metabolic engineering design has been explored, I would expect a wider variety of analysis here.

The results shown for the more general constraint of total enzyme mass are somewhat more consistent and span a variety of effects. However, novelty in the approach here is somewhat more limited compared with existing methods, models and concepts. Thus, I believe the authors need to expand the results section for the full formulation of the method such that this paper can bring the expected level of contribution for MSB.

We thank the reviewer for his very good suggestion. We have included as additional analysis in section 2.4:

- 1. A comparison between the predicted flux distributions by both models, with the help of a customized random sampling method (figure 5A).**
- 2. A comparison of flux distributions to experimental ¹³C flux data from a previous study¹ (table S5, figure S12).**
- 3. A study that shows how flux variability reduction in different pathways relates to total enzyme usage (figure 5C).**

Moreover, in section 2.4 it is essential to know if the experiments with Yeast 6 have been performed with pFBA or FBA. Given the high variability found, I suspect they were performed with FBA and, given the formulation of GECKO, inspired in pFBA, I believe this is not a fair comparison

In the comparison made of FVA results in sections 2.3.1 and 2.4, both the purely metabolic model and the enzyme-constrained model were tested using the same procedure:

- 1) Exchange rates for glucose, growth, oxygen, carbon dioxide, ethanol, glycerol, acetate and pyruvate were fixed at the values predicted by the enzyme-constrained model when minimizing for glucose at a fixed dilution rate, in order to compare the internal flux distribution at the same physiological conditions.**
- 2) Each reaction in the model was first minimized and then maximized to find the flux variability. When doing so, any corresponding reversible reaction was blocked.**

A detailed description of this procedure is available in supplementary material (section 3.2.4). We believe that the comparison is fair, as neither model had the total sum of fluxes minimized. Note here that even though in chemostat conditions we typically minimize total enzyme usage after minimizing glucose consumption (in a similar fashion to pFBA), this was not done in the case of FVA analysis.

As a final remark, after the reviewer's observation, we tested on the purely metabolic model (Yeast7) our FVA analysis using a parsimonious approach, i.e. maintaining the total sum of fluxes at a minimal value as an additional constraint. However, in the case of Yeast 7 this yields only 12 reactions in the whole metabolic network with non-zero variabilities, all of them lower than 0.05 mmol/gDWh. Using enzyme constraints on this already very limited flux space is then not of so much value.

Some minor concerns:

Introduction

Sentence "However, when considering the production of a metabolite of interest, these models typically make the assumption that the uptake rate of the carbon source (e.g., glucose) limits production. This may be an oversimplification because experimental yields are usually considerably lower than the maximum theoretical yields⁵" Although the first sentence is correct, it does not imply the second observation. Usually in Metabolic Engineering design projects the maximum theoretical yield is not assumed.

We agree with the reviewer; this sentence has been therefore rewritten (section 1, 1st paragraph), as such:

"[...] This may be an oversimplification, as metabolic fluxes are limited by their corresponding enzyme levels. However, this cannot be directly tested in traditional GEMs because they do not allow for connecting enzyme concentrations to metabolic fluxes.."

In terms of the present formulation, the authors need to refer to the paper Machado et al 2016 PLOS Comput Biol given the obvious similarities (although overall the GECKO method brings a sufficient level of innovation)

We have now referenced this paper as suggested (section 2.1, 2nd paragraph).

Results

Overall, sections 222 and 223 are of moderate interest but I feel some observations would need to be discussed. As this would probably expand this section, it might make sense to put part of it as supplementary material. As an example: Why proteins in complexes and promiscuous are faster and smaller (for complexes)? Also, the network properties also require further discussion

We think that additional discussion regarding why different types of enzymes are faster and/or smaller would be mostly speculative, and therefore we decided not to address it. Regarding the network properties, even though one could analyze even further the difference between both networks, we believe it would be peripheral and would draw attention away from our main message.

Regarding data in Figure 3E - I miss a comparison with predictions made with nominal maintenance

We have included this analysis in supplementary material (figure S10) and referred to it in the manuscript (section 2.3.2).

Growth rates (233)

For comparison purposes, the authors either use their method with random k_{cat} values or use normal FBA simulations using the uptake rates given by the GECKO simulation. How would the original model behave in FBA in defined medium if all input fluxes are experimentally given (as this is the standard simulation method)? Although for complex medium this might be difficult to get, it is usually obtained in experiments using defined media at least for the carbon source uptake rate. A related question: how do the uptake rates obtained with the new model / formulation compare with the ones observed experimentally?

The studies from which the experimental data was taken in this section^{2,3} are from shake-flask cultivations and do not report the substrate uptake rates, therefore we can only compare the specific growth rates and cannot perform the analysis the reviewer suggests. An exception is one batch performed in a bioreactor under aerobic conditions on glucose³, for which an average biomass yield of 0.12 g/g is reported, which for the specific growth rate reported of 0.4 h⁻¹ corresponds to a specific glucose uptake rate of 19.0 mmol/gDWh. This is in very good agreement with the enzyme-constrained model, which predicts a value of 17.9 mmol/gDWh. Furthermore, we have already shown that both specific uptake and production rates are correctly simulated by our model in chemostats, both under aerobic (figure 3A, 3E and S11) and anaerobic (figure 3D) conditions. Overall, we therefore believe that our enzyme-constrained model correctly predicts specific substrate uptake rates under batch conditions.

Coming back to the first question, if the predicted specific uptake rate by the enzyme-constrained model is a good proxy for the experimental value, then the simulations we show in figure 4A show how the metabolic model would perform with experimental uptake rates, given that the model was limited with all specific uptake rates predicted by the enzyme-constrained model in each condition. We can therefore infer that the purely metabolic model will over-predict growth if we assign experimental uptake rates, as said in the manuscript.

Also in this section, flux values could have been compared with experimental ¹³C data, for example for one fermentable and one non-fermentable carbon source

We have included a detailed comparison of both Yeast7 and ecYeast7 to experimental ¹³C chemostat data in section 2.4 (table S5 and figure S12).

Section 2.5

"Given their unconstrained nature, GEMs tend to overestimate biological performance such as knockout growth and/or production of a specific metabolite of interest." I believe there is a confusion with model and simulation. In this case, other simulation tools (non FBA) such as MOMA or ROOM provide less overestimates for biomass performance.

We agree with the reviewer and hence we have changed the wording of the sentence (section 2.5, 1st paragraph), as such:

“Constrained-based modeling techniques such as FBA tend to overestimate biological performance under perturbed conditions, e.g. knockout growth and/or production of a specific metabolite of interest.”

References

1. Jouhten, P. *et al.* Oxygen dependence of metabolic fluxes and energy generation of *Saccharomyces cerevisiae* CEN.PK113-1A. *BMC Syst. Biol.* (2008). doi:10.1186/1752-0509-2-60
2. Tyson, C. B. & Lord, P. G. Dependency of size of *Saccharomyces cerevisiae* cells on growth rate. *J. Bacteriol.* **138**, 92–98 (1979).
3. Van Dijken, J. P. *et al.* An interlaboratory comparison of physiological and genetic properties of four *Saccharomyces cerevisiae* strains. *Enzyme Microb. Technol.* **26**, 706–714 (2000).

2nd Editorial Decision

14 June 2017

Thank you again for submitting your work to Molecular Systems Biology. I greatly apologize for the delay in getting back to you. Unfortunately one of the reviewers failed to respond to our multiple reminders. Rather than delaying further the process even further, I prefer to make a decision now. As you will see, reviewer #1 is now fully supportive and I am pleased to inform you that we will be able to accept your paper for publication pending the following minor amendments:

- Appendix Figures S1-S15 and Appendix Tables should be called out from the main text as "Appendix Figure S1", "Appendix Figure S2", "Appendix Table S1" etc...
- We would be grateful if you could include a formal "Data and Software Availability Section" after Materials & Methods according to the following generic example:

#Data and software availability

The primary datasets produced in this study are available in the following databases:

- RNA-Seq data: Gene Expression Omnibus GSE46843
(<https://www.ncbi.nlm.nih.gov/geo/query/acc.cgi?acc=GSE46843>)
- Chip-Seq data: Gene Expression Omnibus GSE46748
(<https://www.ncbi.nlm.nih.gov/geo/query/acc.cgi?acc=GSE46748>)
- Protein interaction AP-MS data: PRIDE PXD000208
(<http://www.ebi.ac.uk/pride/archive/projects/PXD000208>)
- Imaging dataset: Dryad Digital Repository doi:10.5061/dryad.35h8v
- Modeling scripts: GitHub <https://github.com/repo>
- [short description of the measurement type]: [full name of the resource] [accession number/identifier] ([resolvable hyperlink])

REFeree REPORT

Reviewer #1:

The authors have responded well to all significant comments in my previous review. Overall, I find this paper to be suitable for publication.

2nd Revision - authors' response

14 June 2017

The authors made the requested changes and submitted the final version of their manuscript.

3rd Editorial Decision

19 June 2017

Thank you again for sending us your revised manuscript. We are now satisfied with the modifications made and I am pleased to inform you that your paper has been accepted for publication.

Corresponding Author Name: Jens Nielsen

Manuscript Number: MSB-16-7411